# Comparative Evaluation of Temporal Modeling in Foundation Models for Echocardiographic LVEF Estimation

**Kimberly Te**[1]                                                    KIMTE@STANFORD.EDU
**Tanveer Syeda-Mahmood**[1]                              TANVEERSYEDA1@STANFORD.EDU
[1] *Stanford University, Stanford, CA, USA*

## Abstract

Automated left ventricular ejection fraction (LVEF) estimation has relied on image-level representations despite echocardiography being an inherently temporal modality. In this study, we conduct a comparative evaluation of four pretrained models covering both image-based representations and video-based representations to study the impact of incorporating video information directly in the estimation of left ventricular ejection fraction. The models (OpenCLIP, EchoCLIP, Echo-Vision-FM, and VideoCLIP) were tested under identical frozen MLP probes across 10 seeds on EchoNet-Dynamic dataset. Our results indicate that VideoCLIP achieved the best 16-frame result (4.79% MAE), while mean-pooled image encoders showed no improvement over single-frame baselines. Reconstruction-based pretraining encoded temporal structure but was not linearly accessible under frozen probing ($R^2 = 0.178$). Within this frozen-probe benchmark, our observed differences in temporal benefit were more consistent with pretraining objective than with input modality alone.

**Keywords:** echocardiography, foundation models, vision language model, video model, ejection fraction, temporal modeling, frozen representation evaluation, contrastive learning

## 1. Introduction

Echocardiography is a temporal imaging modality that visualizes cardiac structure and function across the cardiac cycle. Left ventricular ejection fraction (LVEF), the primary quantitative measure used in heart failure diagnosis and treatment, is defined by the fraction of blood ejected per contraction from ventricular volumes at end-diastole (ED) and end-systole (ES) (Lang et al., 2015). Regression-based approaches have shown that LVEF can be estimated directly from echocardiographic images without intermediate segmentation (Lu et al., 2018), and contrastive foundation models trained on still frames have since achieved strong LVEF performance from frame-based representations (Christensen et al., 2024; Ouyang et al., 2020). This raises a question of whether video-native pretraining provides measurable benefit beyond aggregated frame-level embeddings, which is important for echocardiogram foundation model selection in clinical deployment.

In this study, we do comparative evaluation of state-of-the-art (SOTA) echocardiogram foundation models spanning two design axes, namely, input modality (image vs. video) and pretraining objective (contrastive vs. reconstructive) under a unified frozen-representation evaluation. Three findings have emerged from our evaluation. (1) Mean-pooled frame embeddings provide no temporal benefit over strong single-frame baselines. (2) Reconstruction-based pretraining may encode structure not readily accessible to simple frozen probes. (3) ES frames consistently outperform ED frames, suggesting a practical deployment heuristic. Finally, we conclude that temporal benefit under frozen probing is model-dependent and cannot be assumed from video input modality alone.

## 2. Methods

**Datasets and Models:** LVEF estimation was evaluated on EchoNet-Dynamic (Ouyang et al., 2020), a dataset of 10,030 apical four-chamber (A4C) videos with expert-annotated LVEF and ED/ES frame indices. Of the 160 test samples falling below the 40% treatment threshold, we report the clinically critical misclassification rate (CCMR) as a measure of errors with direct treatment consequence. Since gold-standard EF measurement by the biplane Simpson's method requires both A4C and two-chamber acquisition (2CH) (Lang et al., 2015), we additionally evaluated binary 2CH/4CH view classification on the CAMUS dataset (Leclerc et al., 2019) as a probe of view-discriminative structure (Lu et al., 2018). The SOTA models evaluated were OpenCLIP ViT-B/16 (Cherti et al., 2023) (general-domain image-contrastive, 512-dim), EchoCLIP (Christensen et al., 2024) (cardiac image-contrastive, English echo-report pairs, 512-dim), EVFM (Zhang et al., 2026) (cardiac video-reconstructive, masked autoencoding, 768-dim, 16-frame), and VideoCLIP (Takizawa et al., 2025) (cardiac video-contrastive, Japanese echo-report pairs, 512-dim, 32-frame).

**EF Estimation and View Classification:** A frozen MLP probe was used to test whether clinically predictive structure was linearly accessible in each pretrained representation without encoder adaptation (Alain and Bengio, 2017). Three input conditions were evaluated per model: ED single-frame, ES single-frame, and a 16-frame (16f) clip sampled from a single cardiac cycle. Image models averaged per-frame embeddings for 16f-clips (Christensen et al., 2024). For EVFM, single frames were replicated across all 16 input positions. Its uniform sampling in pretraining contrasts with our 16f clip within cycle approach, which may constrain results. For VideoCLIP, pretrained on 32-frame clips, temporal positional encodings were interpolated to support 16f input. A two-layer MLP probe (256-dim, ReLU, dropout=0.2) was trained identically across models with Adam ($10^{-3}$), early stopping (patience 10), and a maximum of 200 epochs across 10 random seeds. The same frozen-probe protocol was applied to CAMUS for 2CH/4CH view classification. Temporal benefit was defined as the signed difference in mean absolute error (MAE) between 16-frame and best single-frame (negative=improvement).

## 3. Experiments and Results

Table 1 and Figure 1 summarize performance. All six pairwise model comparisons at the 16-frame condition were significant after Bonferroni correction (paired $t$-tests, $\alpha$=0.0083). For image models, mean-pooling provided no temporal benefit (OpenCLIP $\Delta$=+0.04±0.05, EchoCLIP $\Delta$=+0.09±0.04). Both models performed worse than their ES baselines despite domain pretraining substantially improving absolute performance (EchoCLIP ES 5.60% vs. OpenCLIP ES 8.09%). This suggests that simple frame averaging does not recover additional predictive signal beyond the strongest ES frame.

EVFM achieved a frozen $R^2$ of 0.178 and a 16-frame MAE of 8.22% ($\Delta$=−0.26±0.11), falling 4.35 percentage points lower than the fine-tuned result of Zhang et al. (Zhang et al., 2026). This gap is consistent with a probe-accessibility mismatch. VideoCLIP's single-frame performance was modest (8.37% ED, 7.43% ES), but 16-frame input reduced error to 4.79% MAE ($\Delta$=−2.64±0.05), the strongest result in the benchmark. This approaches the 4.1% EchoNet-Dynamic supervised baseline (Ouyang et al., 2020), where Ouyang et al. used end-to-end supervision and beat-by-beat aggregation across multiple cardiac cycles. Causal attribution to video-native pretraining is also limited by three confounds: indirect

Table 1: EchoNet-Dynamic EF regression and CAMUS view classification (mean ± std, 10 seeds). †CCMR descriptive only ($n$=160). ‡Corpus contains EF-labeled reports.

| Model | EF MAE (%)↓ | | | 16-frame | | Temporal | CAMUS | CCMR† |
| | ED | ES | 16f | $R^2$ ↑ | Low-EF↓ | $\Delta$ MAE↓ | Acc (%)↑ | (%)↓ |
| --- | --- | --- | --- | --- | --- | --- | --- | --- |
| OpenCLIP | $8.84_{\pm.04}$ | $8.09_{\pm.03}$ | $8.13_{\pm.03}$ | $.224_{\pm.004}$ | 21.1 | $+0.04_{\pm.05}$ | $89.9_{\pm.5}$ | 11.9 |
| EchoCLIP‡ | $\mathbf{6.78}_{\pm.02}$ | $\mathbf{5.60}_{\pm.02}$ | $5.69_{\pm.03}$ | $.610_{\pm.004}$ | 11.3 | $+0.09_{\pm.04}$ | $90.9_{\pm.7}$ | 7.7 |
| EVFM | $8.96_{\pm.05}$ | $8.48_{\pm.09}$ | $8.22_{\pm.07}$ | $.178_{\pm.011}$ | 22.1 | $-0.26_{\pm.11}$ | $81.5_{\pm3.0}$ | 12.5 |
| VideoCLIP‡ | $8.37_{\pm.04}$ | $7.43_{\pm.02}$ | $\mathbf{4.79}_{\pm.03}$ | $\mathbf{.738}_{\pm.004}$ | $\mathbf{7.4}$ | $-2.64_{\pm.05}$ | $\mathbf{91.8}_{\pm.6}$ | $\mathbf{4.8}$ |

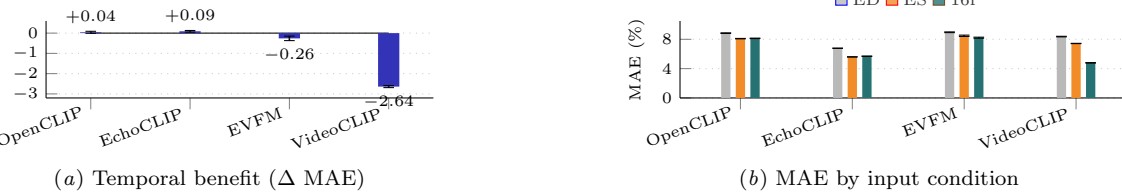

($a$) Temporal benefit ($\Delta$ MAE)  ($b$) MAE by input condition

Figure 1: (a) Temporal benefit ($\Delta$ MAE) per model. (b) MAE by input condition; ES outperforms ED across all models.

EF supervision in the echo-report pretraining corpus of both VideoCLIP and EchoCLIP, positional encoding interpolation from 32 to 16 frames, and unverifiable proximity of the Japanese training set to EchoNet-Dynamic. Across all four models, ES frames outperformed ED by 0.48–1.18 percentage points (Figure 1b), suggesting a consistent deployment heuristic without retraining. On CAMUS 2CH/4CH classification, all models except EVFM exceeded 89% accuracy, EVFM reached 81.5% with notably higher seed variance ($\pm3.0$ vs. $\pm0.5$–0.7), consistent with reconstruction-based representations being less stably accessible on discrete geometric tasks. The CCMR and low-EF MAE both followed the global ranking: VideoCLIP lowest (4.8%, 7.4%) and EVFM highest (12.5%, 22.1%).

## 4. Discussion and Conclusion

Within this frozen-probe evaluation, temporal benefit was more closely tied to representation accessibility than to input modality alone. The null temporal result for the image models suggests that clinically predictive structure was already concentrated in the strongest single frame, notably at ES, and was not improved by mean-pooling additional frames. EVFM provides a useful contrast. Despite being video-native, it showed only modest frozen-probe gains, and its weaker performance extended to CAMUS, where higher seed variance suggests less stable access to view-discriminative structure under a simple probe. EchoCLIP and VideoCLIP share the same potential EF leakage confound but showed opposite temporal behavior, suggesting that leakage alone is unlikely to explain VideoCLIP's large multi-frame gain. A practical implication is that ES frame selection generalized across all four models and requires no retraining. These findings suggest that pretraining objective should be treated as a first-order design consideration when selecting or adapting foundation models for clinical echocardiographic tasks, particularly in deployment settings where full fine-tuning is unavailable.

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
