# OpenReview forum: "Comparative Evaluation of Temporal Modeling in  Foundation Models for Echocardiographic Left Ventricular Ejection Fraction Estimation"
_MIDL.io/2026/Short_Papers — MIDL 2026 - Short Papers Poster_

### Official Review · Reviewer_nvHX · 2026-04-26
**Comparative Evaluation of Temporal Modeling in Foundation Models for Echocardiographic LVEF Estimation**

**Rating:** 4
**Confidence:** 5

**Review:**

This work addresses a relevant and timely problem, namely the evaluation of pretrained models for echocardiographic video analysis, with potential implications for clinical practice. The study is well-motivated and provides a clear comparative framework across multiple architectures and tasks, which strengthens its practical relevance.

In terms of quality, the experimental section is solid and supports meaningful conclusions, particularly regarding the benefit of incorporating temporal information. The clarity of the manuscript is generally good, with well-presented results and conclusions.

The originality of the work remains moderate, as it primarily focuses on the comparative evaluation of existing methods. In addition, some methodological aspects, such as the pretraining of the MLP and the adaptations applied to the models, are not sufficiently studied or may impact the fairness of the comparison.


Pros:

- Relevant topic with clear clinical implications.
- Comprehensive evaluation across multiple models, tasks, and temporal configurations.
- Use of publicly available datasets supporting reproducibility.
- Clear and well-supported conclusions.

Cons:

- Limited originality, as the contribution is mainly comparative.
- Insufficient details regarding some methodological components (e.g., MLP pretraining).
- Potential bias in the comparison due to model adaptations not reflecting optimal configurations.

**Summary:**

This study presents a comparative evaluation of four pretrained models, covering both image-based and video-based representations, to assess the impact of incorporating temporal information for ejection fraction estimation and view classification. All models are adapted to process the same input data, enabling a fair comparison. Two datasets were used: EchoNet-Dynamic for the estimation of ejection fraction and CAMUS for view classification. The results provide several clear insights. In particular, the best-performing method, VideoCLIP, achieves a mean absolute error (MAE) of 4.8% for ejection fraction estimation and an accurcy of 91.8 for view classification.

**Strengths:**

- The relevance of the topic, as the automatic and robust estimation of ejection fraction from echocardiographic videos could have a significant impact on clinical practice.
- The evaluation of the models on two distinct downstream tasks.
- The use of publicly available datasets, which supports the reproducibility of the experiments.
- A solid experimental section, involving four state-of-the-art methods evaluated under different temporal input configurations.
- The clarity of the conclusions.

**Weaknesses:**

- The description of the MLP pretraining is insufficient and should be further detailed, particularly regarding the data used (origin and dataset size).
- The adaptations applied to the models to match the experimental setup (e.g., input duplication, interpolation of temporal positional encodings) prevent a fair assessment of each method in its optimal configuration. This may affect both the reliability of the reported results and the validity of the conclusions.

**Justification Of Rating:**

The paper addresses a relevant problem and provides a solid comparative evaluation with clear and practically meaningful conclusions. In particular, the study offers useful insights into the role of temporal information in echocardiographic analysis. However, the contribution remains moderately original, and some methodological aspects (e.g., MLP pretraining and model adaptations) are not sufficiently detailed or studied, which may affect the fairness of the comparison. Overall, despite these limitations, the strengths of the experimental evaluation and the practical relevance of the work support a weak accept.

---

### Decision · Program_Chairs · 2026-05-08

Accept (Poster)